# Transcranial Magnetic Stimulation as a Tool to Promote Smoking Cessation and Decrease Drug and Alcohol Use

**DOI:** 10.3390/brainsci13071072

**Published:** 2023-07-14

**Authors:** Tal Harmelech, Colleen A. Hanlon, Aron Tendler

**Affiliations:** 1BrainsWay Ltd., Winston-Salem, NC 27106, USA; 2Wake Forest School of Medicine, Winston-Salem, NC 27106, USA; 3Department of Life Sciences, Ben Gurion University of the Negev, Beer-Sheva 84105, Israel

**Keywords:** smoking cessation, nicotine use disorder, cocaine use disorder, alcohol use disorder, substance use disorder, addiction, dependance, transcranial magnetic stimulation, substance abuse, relapse, neural circuits, noninvasive intervention

## Abstract

Repetitive transcranial magnetic stimulation (rTMS) is a noninvasive, drug-free, neural-circuit-based therapeutic tool that was recently cleared by the United States Food and Drug Associate for the treatment of smoking cessation. TMS has been investigated as a tool to reduce consumption and craving for many other substance use disorders (SUDs). This review starts with a discussion of neural networks involved in the addiction process. It then provides a framework for the therapeutic efficacy of TMS describing the role of executive control circuits, default mode, and salience circuits as putative targets for neuromodulation (via targeting the DLPFC, MPFC, cingulate, and insula bilaterally). A series of the largest studies of TMS in SUDs are listed and discussed in the context of this framework. Our review concludes with an assessment of the current state of knowledge regarding the use of rTMS as a therapeutic tool in reducing drug, alcohol, and nicotine use and identifies gaps in the literature that need to be addressed in future studies. Namely, while the presumed mechanism through which TMS exerts its effects is by modulating the functional connectivity circuits involved in executive control and salience of drug-related cues, it is also possible that TMS has direct effects on subcortical dopamine, a hypothesis that could be explored in greater detail with PET imaging.

## 1. Introduction

Substance use disorders (SUDs) are a major contributor to morbidity and mortality worldwide [1]. While there are pharmacological treatments available for some SUDs, others do not have any established therapies (e.g., methamphetamine, cocaine). Furthermore, short-term relapse rates are often as high as 60% across multiple classes of drugs [1]. One major shortcoming of pharmacologic and behavioral treatments for SUDs is that these approaches do not directly modulate relevant neural circuits. These include circuits that contribute to the cycle of chronic use, remorse, abstinence, and cue- or stress-induced relapse that is common to multiple classes of abused substances.

To make critical strides in treating drug and alcohol use disorders, it is imperative we add noninvasive neural-circuit-based interventions to the toolbox of options for individuals seeking to reduce or eliminate their reliance on substances [2,3,4]. In this context, a recent consensus paper outlined the recommended criteria for brain stimulation in SUDs and called for more multicenter studies [5]. This paper reviews the current state of knowledge regarding the use of repetitive transcranial magnetic stimulation (TMS) as a therapeutic tool to reduce drug, alcohol, and nicotine use.

## 2. Neural Circuits Guiding Drug-Taking Behavior

SUDs are chronic, recurring diseases. The DSM-V recognized 10 separate classes of abused substances: alcohol, caffeine, cannabis, hallucinogens, inhalants, opioids, sedatives, anxiolytics, stimulants, and tobacco. While this is a long (and some would say “incomplete”) list, there are several core features that unite the addiction process. From a behavioral perspective, these include the persistent, compulsive use of the drug despite attempts to cut back, cue- or stress-induced craving, as well as a physiologic tolerance and withdrawal state. These features often lead to impairment in performing daily responsibilities and maintaining relationships.

From a biological perspective, the dopaminergic system is the most studied aspect of addictive behavior [6]. Positron emission tomography (PET) has demonstrated, for example, that patients with SUDs have reduced ventral striatal D2 receptors and diminished dopamine release [7]. While a full review of this research is beyond the scope of this article, it is important to mention that the dopamine hypothesis of drug addiction [8] was developed by basic science researchers and later translated into human studies [9,10].

The biological mechanisms guiding initiation, dependence, and relapse to drugs, however, are not limited to the dopamine-rich areas of the striatum. The cortical areas that have afferent and efferent connectivity with the striatum are also critical to cue-induced craving and compulsive drug taking. The executive control network, including the dorsolateral PFC (DLPFC), the posterior parietal cortex, and the dorsal cingulate cortex, governs and regulates action patterns, decision making, and self-control. In addition, the ventral PFC network, including the medial PFC (mPFC), the orbitofrontal cortex (OFC), the insular cortex, and the ventral anterior cingulate cortex, modulate limbic arousal and emotion processing [11]. An imbalance of these two systems is thought to contribute to the vulnerability to develop and relapse into SUDs [12]. Additionally, using functional MRI, Jousta and colleagues (2022) demonstrated that connectivity to many of the aforementioned areas is a transdiagnostic feature of SUDs [13]. Considered together, this body of work indicates that there are many potential therapeutic targets for rTMS in the SUD field.

## 3. Developing Transcranial Magnetic Stimulation as a Neural-Circuit-Based Treatment for Addiction

The earliest treatments for addiction sought to address the underlying maladaptive behaviors, including compulsive use despite negative consequences. Behavioral interventions are currently the mainstay of treatment in outpatient substance abuse treatment programs. These behavioral interventions are likely crucial for meaningful lifestyle changes in patients, but their efficacy alone is relatively low and likely to be heightened with an adjuvant biologic intervention.

With accumulating neuroimaging evidence that irregularities in these brain circuits contribute to chronic use and relapse and the introduction of transcranial magnetic stimulation (TMS) as the first noninvasive neural-circuit-based intervention in psychiatry, we are at a critical juncture in the history of developing an evidence-based neural circuit therapeutic for a variety of SUDs.

The remainder of this review will discuss TMS as an emerging noninvasive therapeutic option for individuals with a variety of drug and alcohol use disorders. The interest in developing TMS as a tool for addiction treatment was undoubtedly magnified by the influential findings of Strafella and colleagues (2001), wherein they demonstrated that there is a causal relationship between TMS of the prefrontal cortex and dopamine binding in the caudate nucleus [14]. This was further strengthened by the results from Zangen and colleagues (2002) demonstrating that TMS increases extracellular dopamine and glutamate in the ventral striatum [15]. A full review of the effects of TMS on glucose, cerebral blood flow, and dopamine in the brain is presented by Kinney and Hanlon (2022) [16].

The largest study of TMS as a therapeutic tool for substance use disorders was completed in 2020 and led to the FDA clearance of the Deep TMS H4 coil as a tool for smoking cessation [17]. Following a review of this FDA-pivotal trial, we will discuss a select set of studies that have been conducted using a variety of TMS techniques in other SUDs. A list of published studies can be found in Appendix A.

## 4. Extant Literature of TMS for Substance Use Disorders

### Smoking Cessation

Smoking is the leading preventable cause of mortality and morbidity in the United States, contributing to approximately 443,000 deaths annually [18]. Strikingly, this is more than the deaths attributable to alcohol, illicit drug use, homicide, AIDS, and suicide combined. Furthermore, the combined health and loss-of-productivity costs associated with smoking are substantial, exceeding USD 300 billion per year [19], or nearly 10 times NIH’s entire 2016 budget. In 2015, approximately 36.5 million Americans (15%) were regular smokers. Of these, 68% wanted to quit smoking [20]. Yet, only 6% who attempt to quit without assistance maintain abstinence for 30 days [21]. Current smoking cessation treatments (e.g., nicotine replacement therapy, cognitive behavioral therapy [CBT], and non-nicotine medications) have a success rate of approximately 30% [22].

This multisite clinical trial was the result of several decades of work in both preclinical research laboratories and small clinical trials using a variety of TMS coils [23,24,25,26,27,28,29,30,31,32,33,34,35,36,37,38,39]. In 2020, the United States Food and Drug Administration (FDA) cleared the H4 coil for use as an aid in short-term smoking cessation for adults (Deep TMS, BrainsWay, Jerusalem, Israel). This decision came as a result of a double-blind, randomized, sham-controlled (RCT) multisite clinical trial [17]. This trial used a TMS coil that modulated the MPFC and the insula bilaterally, both of which are evidence-based targets for intervention [11,13]. Currently, the H coils (e.g., H1, which is FDA-cleared for depression; H4, which is FDA-cleared for smoking cessation) are the only TMS coils available to stimulate the left and right sides of the brain simultaneously. Consequently, they may be uniquely suited to disorders that involve the executive control network (DLPFC) and the salience network (insula, ACC) which are both bilateral in nature.

This positive decision by the FDA is a major milestone in the field of substance use treatment, and the result of several decades of work from basic science laboratories, neuroimaging laboratories, and clinical research studies evaluating many types of TMS coils on multiple cortical targets. A full list of studies on TMS in smoking is provided in Appendix A.

While the majority of early studies evaluated focal rTMS over the left dorsolateral PFC (DLPFC), executive control and limbic activity are bilateral in nature. It is important to develop a TMS strategy that could target both the medial PFC and lateral PFC bilaterally. In one study, 13 active TMS sessions with H4 (20 min/weekday for three weeks) led to a higher one-month quit rate and reduced cigarette consumption compared with the sham procedure [34].

These positive results led to a prospective, multicenter RCT [17] that included 262 chronic smokers meeting DSM-5 criteria for tobacco use disorder (TUD). All participants failed at least one attempt to quit, with 68% failing at least three attempts. The key exclusion criteria included current treatment for smoking and any other psychiatric disorder diagnosed according to the DSM-5. They received three weeks of daily active or sham TMS to the lateral prefrontal and insular cortices with the H4 coil, followed by once-weekly sessions for three weeks. Participants were instructed to refrain from smoking for at least two hours prior to each visit. Each TMS session was preceded by a 5 min provocation procedure that included participants imagining their greatest trigger for craving, listening to an audio script with instructions to handle a cigarette and a lighter, and viewing pictures of smoking. Immediately following the provocation and rMT determination, the helmet was aligned symmetrically and moved 6 cm anteriorly. The intensity of the stimulator was set to 120% of the minimal motor threshold. Then, 60 trains of 30 pulses (i.e., a total of 1800 pulses) were applied at 10 Hz (3 s each train) with 50 s inter-train intervals.

The primary endpoint was a four-week continuous quit rate (CQR) in the intent-to-treat (ITT) group based on participants’ self-reports and verified with urine cotinine levels for up to 18 weeks. In the ITT analysis set (N = 234), the CQR until week 18 was 19.4% following active and 8.7% following sham TMS (X^2^ = 5.655, *p* = 0.017; see Figure 1A). Among completers (N = 169), the CQR until week 18 was 28% and 11.7%, respectively (X^2^ = 7.219, *p* = 0.007; see Figure 1A). The reduction in cigarette consumption and craving was significantly greater in the active than in the sham group as early as two weeks into treatment (see Figure 1B). Adverse events were typical to those of similar TMS devices and were at least comparable to those of medications. The drop-out rate (at week 6) was 39% for the active group and 32% for the sham group, without a significant difference between groups. This study established a safe treatment protocol that promotes smoking cessation by stimulating the medial, ventrolateral, and insula cortices. Based on this multisite study, the H4 coil received FDA clearance for short-term smoking cessation for adults.

The data from this multicenter trial also characterized demographic and smoking history factors that moderated TMS efficacy [38]. Thus, participants younger than 40 had four times the quit rate than those older than 40. Additionally, subjects who quit following treatment smoked 10 years less than nonquitters. Moreover, patients with a mother who did not smoke were twice as likely to quit than those who grew up with a mother that smoked. Strikingly, patients with more than 12 years of education had 7 times the quit rate than patients with less education (see Figure 2). The authors hypothesized that participants with less education and more extensive smoking histories need a longer treatment course and/or combined treatment modalities. This is possibly related to the challenges of inducing neuronal modifications in those with greater physical and psychological dependence.

Another mechanistic, proof-of-concept study with the H4 coil revealed that participants assigned to active TMS were slower to initiate smoking their first cigarette than the sham group, an observation consistent with smoking disruption [39]. The neuroimaging data collected in this experiment showed an overall decrease in insula activity and changes in the resting-state connectivity between the insula and the default mode network (DMN) following active TMS. Research has shown that the insula plays a crucial role in craving. This has been demonstrated in stroke patients, where damage to the insula results in a significant reduction in cigarette cravings, withdrawal symptoms, and nicotine-seeking behavior [40]. The insula is an essential component of the brain’s salience network, and abnormalities in its connectivity have been linked to smoking behavior. For instance, enhanced connectivity between the insula and the default mode network (DMN) and salience network has been associated with smoking withdrawal [41], while increased connectivity between the insula and a larger salience network has been correlated with cigarette cravings [42] and fMRI activation in response to smoking cues [43].

## 5. What Does the FDA-Cleared H4 Coil for Smoking Cessation Modulate?

Figure 2 displays the electric field distribution maps for the H4 coil. Fiocchi et al. quantified the electric field distribution and depth induced using an H4 coil and a Figure-8 coil [44]. The H4 coil induced the highest electric fields in the PFC, insula, and ACC. Moreover, it can induce amplitude (E) in the deepest tissues, ranging between 20% and 40% of the cortical peak, and it can penetrate the cortex up to 4 cm with E > 50% of the cortical peak.

### 5.1. Alcohol Use Disorder

Despite major advances in our understanding of the neural circuits that contribute to alcohol use disorder (AUD), there are still no approved neural-circuit-based therapeutics. Furthermore, the involvement of multiple neurotransmitter mechanisms represents a challenge to developing novel pharmacotherapies. An attractive alternative or complementary strategy is to noninvasively target brain circuit activity associated with pathology, rather than individual neurochemical systems. rTMS offers opportunities for this type of noninvasive, network-targeting intervention. In this context, the majority of rTMS studies for AUD have targeted the left DLPFC with mixed success ([45,46,47,48,49,50,51,52,53,54,55], see Appendix A).

The ACC and mPFC may offer mechanistically attractive candidate treatment targets for rTMS. These include their interactions with other cortical and subcortical areas subserving craving, reward-related decision making, and top-down control of drug-seeking behavior [56,57,58,59,60,61,62]. For example, ACC neural responses to alcohol-related cues are associated with self-reported craving, addiction severity, and relapse [56,57,58].

The two biggest TMS studies in AUD to date were published in 2021 [63] and 2022 [64]. While their sample sizes were modest, the complementary nature of their results is a unifying victory for the future of TMS as a tool for AUD. These two studies were carried out on different continents, by two independent research groups, using two different TMS coils. The studies had many unique things in common, however. They both focused on the medial wall (rather than the DLPFC, which had been the most common target in the past); both used alcohol cue provocation before the TMS treatment; delivered a relatively high dose of TMS each session (demonstrating feasibility in this population); followed the patients for 3 months; and utilized both self-report and urine-based assessments of alcohol usage.

Harel and colleagues (2021) [63] targeted the dorsal MPFC/cingulate cortex using an H7 coil (10 Hz, 3000 pulses per session, 15 sessions delivered 5 times per week). McCalley and colleagues (2022) [64] targeted the ventral MPFC (frontal pole) using a Figure-8 coil (cTBS, 3600 pulses per session, 10 sessions delivered 3 times/week). This protocol was based on an earlier pilot study [65]. In both studies, alcohol consumption was lower in the group that received active relative to sham TMS for 3 months (the longest timepoint evaluated; see Figure 3). In both studies, active TMS resulted in lower functional connectivity between the mPFC and other key addiction-related brain regions, including the subgenual ACC [63] (see Figure 4), the striatum [64], and the mPFC–insula [64].

Considered together, these data support the rationale for a full-scale confirmatory multicenter trial. The therapeutic benefits of TMS appear to be associated with persistent changes in brain network activity. It is also important that TMS be accompanied by behavioral provocation [55], as this likely controls the state that the brain is in during TMS delivery, which may make it more amenable to change based on reconsolidation theory [66].

### 5.2. Cocaine Use Disorder

Brain stimulation techniques such as rTMS may be potential therapies for cocaine use disorder (CUD) [67]. For example, previous studies reported that TMS to the DLPFC reduced craving for cocaine ([67,68,69,70], see Appendix A). One study reported that two sessions of high-frequency TMS applied to the right DLPFC reduced craving, although the effect dissipated after 4 h [68]. Another study showed that rTMS, applied to the left DLPFC, had no acute effect on craving but instead gradually reduced it over 10 daily sessions [69]. A third study used TMS directed bilaterally at the DLPFC (H1 coil), as an add-on treatment in the clinic for CUD. The authors reported that craving gradually decreased over a month of treatment [70]. Martinez et al. [71] investigated the effect of TMS on cocaine self-administration in the laboratory. In these sessions, CUD participants chose between cocaine and an alternative reinforcer (i.e., money) to directly measure cocaine-seeking behavior. The authors measured craving but saw no effect with TMS directed at the mPFC and ACC or compared with the sham group. There were, however, large between-group differences at baseline prior to receiving TMS, making it hard to interpret the data.

More recent small-scale clinical trials indicate that high-frequency TMS to the PFC may improve abstinence from cocaine ([72,73], see Appendix A). Terraneo et al. [72] randomized 32 CUD subjects to receive high-frequency TMS (15 Hz) to the DLPFC or a control consisting of pharmacological treatment. The results revealed that the TMS group had more cocaine-free urine drug tests than the control group. Bolloni et al. [73] reported similar results in CUD subjects treated for 1 month with high-frequency (10 Hz) TMS (H1 Coil) or a sham procedure applied bilaterally over the DLPFC. They found a reduction in cocaine use, measured with hair analysis, in the active TMS group at 3 months (*p* = 0.02) and 6 months (*p* = 0.01) post-treatment. Martinez et al. [71] found a significant group–time effect (*p* = 0.02), where the choices for cocaine decreased between sessions 2 and 3 in the high-frequency TMS group but not in the low-frequency or sham groups. The results of this study complement previous findings and demonstrate that the stimulation of the cortex with a wide, deep electric field may be particularly useful for cocaine use disorders. As with obsessive compulsive disorder [74] most studies in this field are pairing the MS with a behavioral prime or provocation.

A pilot study by Sanna et al. [75] investigated the efficacy of intermittent TBS (iTBS) compared with 15 Hz stimulation over the bilateral PFC and insula using the H4 coil. They reported that the effect of iTBS on cocaine consumption and craving was comparable to 15 Hz stimulation. Both treatments had low dropout rates and similar safety and tolerability profiles. The same group explored the long-term outcome of iTBS with the H4 coil in CUD and the possible influence of maintenance treatments on abstinence and drop-out rates [76]. In a retrospective analysis of 89 patients who were exposed to 20 sessions of iTBS, 61 (81%) patients were abstinent at the end of treatment and were followed up for 12 months. Among these, 27 patients chose to follow a maintenance treatment (M) for 3 months, whereas 34 patients chose not to adhere to a maintenance treatment (NM). Overall, among patients reaching the 12-month follow-up endpoint, 69.7% were still abstinent, and 30.3% relapsed. The drop-out rate was significantly higher in NM patients than in M patients (58.82 vs. 29.63%; *p* = 0.04). These observations demonstrate the long-term therapeutic effect of bilateral PFC iTBS in decreasing cocaine consumption. Moreover, they underscore the importance of continuation treatments to consolidate abstinence and decrease drop-out rates over time.

### 5.3. Opioid Use Disorders

In 2012, Taylor and colleagues published an influential paper demonstrating that 10 Hz TMS to the left DLPFC reduces perceived pain in healthy volunteers, but this effect can be blocked by naloxone, an opiate antagonist [77]. This suggests that TMS can induce analgesia via endogenous opioid release. This set the stage for a growing enthusiasm regarding the link between TMS, pain, and opiates, as well as curiosity about the use of TMS as a tool to help individuals with opioid use disorder. Shen and colleagues (2016) demonstrated that a single session of active TMS (10 Hz, 100% motor threshold, left DLPFC) decreased subjective craving and that this effect continued after an additional five sessions of TMS (relative to sham) [78]. Beyond the DLPFC however, there is some interest in the motor cortex as a potential treatment target. In 2021, for example, Imperatore and colleagues demonstrated that 10 daily sessions of 10 Hz TMS to the motor cortex (90% motor threshold) had larger effects on pain (Cohen’s d: 0.7) and urge-to-use opiates (Cohen’s d: 0.5) than left DLPFC stimulation (110% motor threshold) [79].

### 5.4. Cannabis Use Disorder

Another area of emerging interest is the application of TMS as a therapeutic tool in cannabis use disorder. With expanding availability and momentum toward the legalization of marijuana in the United States, the number of individuals with cannabis use disorder has been steadily increasing [80]. In contrast to other SUDs, however, very little research has been conducted on the efficacy of TMS as a tool for cannabis use disorder. The first experimental study was a double-blind crossover study that evaluated the effect of a single session of TMS on event-related brain reactivity to cues [81]. This was followed by a second experimental study evaluating the feasibility and tolerability of TMS to the left DLPFC in 18 individuals with cannabis use disorder [82].

Sahlem and colleagues (2020) followed this with a publication evaluating the safety and tolerability of 20 sessions of 10 Hz TMS (2/day, 10 days) [83]. They found that of the nine participants enrolled, only three were able to complete the protocol as designed. Why might this be? The authors eloquently stated that many of these individuals have external responsibilities that do not allow them to come in every workday for treatment. This is a very important finding and one that should inform future treatment design in the SUD field in general.

Consequently, the research team designed a new two-site, randomized, double-blind, sham-controlled study evaluating the efficacy of TMS as a tool to decrease craving (Marijuana Craving Questionnaire Short-Form (MCQ-SF)) as well as cannabis use (10 Hz, 20 sessions; 2 sessions/day, 2 days/week, 5 weeks; 4-week follow up). Of the 72 individuals enrolled, the individuals who received active TMS reported more weeks of abstinence and fewer days per week of cannabis use in the follow-up period [84]. This is one of the largest studies that has been carried out in the field of TMS for SUDs and will hopefully lead to larger, multisite clinical trials to evaluate TMS as a potential therapeutic tool for this vulnerable population, currently underserved by pharmacotherapeutic options.

## 6. Discussion

As knowledge about the neural networks involved in maintaining and breaking the cycle of drug use, abstinence, and relapse continues to develop, there is growing enthusiasm for a neural-circuit-based intervention for treating SUDs. TMS is one of the most promising treatments given its ability to modulate the neural circuits involved in cue reactivity and executive control [2,3,4,5,85,86,87,88,89], as well as the recent approval of a unique form of TMS for short-term smoking cessation treatment. Multiple converging lines of evidence support the use of TMS for alcohol use disorder and cocaine use disorder. To date, most clinical research studies have focused on increasing activity in the executive control network either unilaterally (Figure-8 coils) or bilaterally (Deep TMS H coils) modulating the DLPFC. Recently, there has been growing interest in modulating the medial prefrontal cortex, a core structure of the default mode network, and the cingulate cortex and bilateral insula, the core regions in the salience network [87].

In this paper, we reviewed the currently available evidence for TMS efficacy in each SUD category. In TUD, there are 13 studies (mostly double-blind, RCTs) utilizing Figure-8 coils, 12 of which targeted the DLPFC. While these studies demonstrated significant short-term effects on smoking-related behavior, no persistent quitting or sustained abstinence was demonstrated. Three RCTs utilizing TMS H4 coils targeted the lateral prefrontal and insular cortices. The first demonstrated sustained abstinence (33% abstinence rate 6 months post-treatment), and the follow-up multicenter study led to the first FDA clearance of a TMS device as an aid in short-term smoking cessation. Post hoc analyses revealed age, education, and smoking history to be moderators of treatment efficacy. Electric field modeling also demonstrates that the H4 coil modulates the MPFC and VLPFC/insula bilaterally, which are critical nodes of the default mode network and the salience network. The engagement of these canonical networks with wide fields may be the underlying reason for the positive outcomes of these trials.

In AUD, 14 studies have been conducted, most of which targeted the DLPFC (as in smoking). The results of DLPFC stimulation studies are mixed. The largest double-blind sham-controlled clinical studies to date modulated the medial prefrontal cortex in different ways, and both had positive effects on alcohol consumption 3 months after TMS. Harel and colleagues (2021) [63] targeted the dorsal MPFC and the cingulate cortex using an H7 coil (10 Hz, 15 sessions), while McCalley and colleagues (2022) [64] targeted the ventral MPFC (frontal pole) using a Figure-8 coil (cTBS, 10 sessions). Both studies demonstrated that active TMS reduced alcohol consumption relative to sham TMS for up to 3 months (which was the longest timepoint evaluated). Both studies also demonstrated that active TMS resulted in lower functional connectivity between the mPFC (a transdiagnostically relevant brain region involved in cue reactivity) and other key addiction-related brain regions, including the subgenual ACC [63], the striatum [64], and the mPFC–insula [64]. In order to take these promising findings to the next level, however, it is critical to secure the funding and support for a multicenter trial.

Beyond tobacco cessation and AUD, there is also a growing body of research on TMS for cocaine use disorder (especially in Europe, where it already has a CE mark), methamphetamine use disorder, and to a lesser extent, opioid dependence and cannabis use disorder. To investigate the overall effects of TMS on craving and substance consumption, Zhang et al. [88] conducted a systematic review and meta-analysis of 26 RCTs (n = 748) published from January 2000 to October 2018 in patients with nicotine, alcohol, and illicit drug dependence. The results showed that high-frequency TMS of the left DLPFC significantly reduced craving (Hedges’ g = −0.62; 95% CI, −0.89 to −0.35; *p*< 0.0001), compared with sham stimulation. Moreover, meta-regression revealed a significant positive association between the total number of stimulation pulses and effect size among studies using excitatory left DLPFC stimulation (*p*= 0.01). The effects of other rTMS protocols on craving were not significant. However, when examining substance consumption, excitatory TMS to the left DLPFC and excitatory Deep TMS of the bilateral PFC and insula revealed significant consumption-reducing effects, compared with sham stimulation.

Another systematic review summarizing evidence of TMS in SUDs suggested that there are several features of the studies that have had positive results for SUDs [89]. Specifically, successful treatment strategies include several minutes of detailed craving induction before stimulation, which activate the networks relevant to the treatment through Hebbian principles in the form of paired associative stimulation. The degree of engagement and dampening of acute cravings may be predictive of long-term treatment success. Finally, there appears to be a brief period of increased plasticity post-stimulation that can be exploited to reinforce learning by way of personalized motivational statements. One of the most recent studies in the cocaine use field highlighted the importance of a maintenance protocol for the sake of evaluating long-term outcomes in this population [76]. The durability of effects is most important for successful addiction treatment, and future studies should ideally conduct follow-up measurements after the completion of the acute TMS intervention.

As mentioned at the beginning of this review, the dopamine hypothesis of addiction has been a guiding force in the field of addiction [8,9,10] and is still a key biological target of multiple novel pharmacotherapeutics for addiction. When Strafella and colleagues (2001) demonstrated a causal relationship between TMS of the prefrontal cortex and dopamine binding in the caudate nucleus [14], a new world of device-based therapeutic opportunities for SUDs unfolded. Unfortunately, despite a wealth of knowledge about the effects of TMS on dopamine, glucose, and cerebral blood flow and dopamine [16], very little research in the TMS for the SUD field has directly investigated the effects of TMS protocols on the neurotransmitters involved in the addiction process. While PET studies are expensive and expose participants to radioactive tracers, there are novel tracers available to investigate dopamine, opioids, nicotinic acetylcholine receptors, and GABA, all of which are underutilized resources that could add a significant amount of information to the field of neuromodulation-based therapeutics for SUD—either through basic science studies or clinical research.

## 7. Key Points

-The executive control network, the default mode network, and salience networks are all possible targets for TMS treatment for SUD;-These networks can be modulated unilaterally with Figure-8 coils and bilaterally with H coils;-The H4 coil, which modulates the MPFC and the VLPFC/insula bilaterally, was FDA-cleared for smoking cessation;-The connectivity profile of lesions disrupting smoking is similar to that of lesions reducing the risk of alcoholism, which suggests the potential transdiagnostic relevance of a TMS coil targeting these regions;-Behavioral provocation appears to enhance the effects of TMS.

## Figures and Tables

**Figure 1 brainsci-13-01072-f001:**
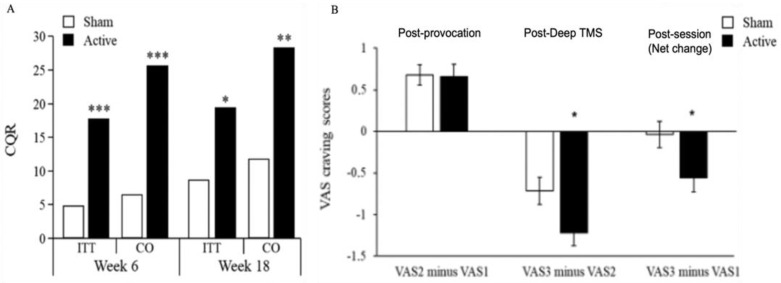
Adapted from [17]: (**A**) Four-week continuous quit rate (CQR) until week 6 and week 18 in patients receiving active or sham TMS. Only participants who were abstinent at week 6 were followed up to week 18. The analysis is presented in the full intent-to-treat (IIT) sample as well as the per-protocol group (which includes the individuals in the IIT sample who finished all TMS sessions). The per-protocol group is listed here as completers (CO). * *p* < 0.05, ** *p* < 0.01, *** *p* < 0.001. (**B**) Acute changes in Visual Analogue Scale (VAS) craving scores following provocation and following Deep TMS in patients receiving active or sham treatment (first session). Overall changes in craving during the first session indicate that craving in the sham group returns to baseline, whereas it is reduced compared with baseline in the active group (*p* = 0.026). * *p* < 0.05.

**Figure 2 brainsci-13-01072-f002:**
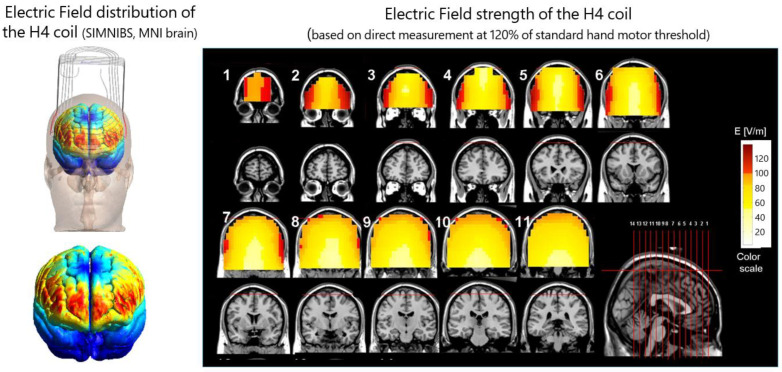
Distribution of electric fields induced by the H4 coil. The spatial topography of the superficial cortical electric field from the H4 coil is shown on a standard brain (MNI template, SIMNIBS software; **left**). Th windings of the coil are indicated by gray lines. The electric field distribution was also measured in a phantom model of the human head (15 × 13 × 18 cm), filled with physiologic saline solution (**right**). The colored field maps indicate the electrical field absolute magnitude in each pixel, for 10 coronal slices, 1 cm apart, along with the appropriate MRI coronal images. The red colors indicate field magnitude above the threshold for neuronal activation, which was set to 100 V/m. The H4 field maps were adjusted for 120% of the hand motor threshold in accordance with FDA-cleared protocols.

**Figure 3 brainsci-13-01072-f003:**
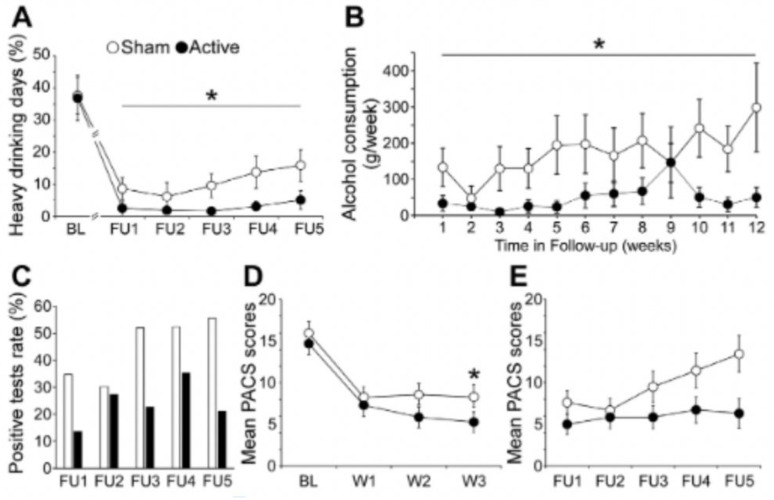
Adapted from [63]. Alcohol consumption and craving. Follow-up times: FU1, 1 week; FU2, 2 weeks; FU3, 4 weeks; FU4, 8 weeks; and FU5, 12 weeks: (**A**) Percentage of heavy drinking days during the follow-up phase showed significant main effects of group (*p* = 0.037, mean difference 7.7%, Cohen’s d = 0.5). (**B**) Alcohol consumption during the follow-up phase showed a significant main effect of groups (*p* = 0.02, mean difference = 121.78 g, Cohen’s d = 0.47). (**C**) Percentages of positive urine ethyl glucuronide samples during the follow-up visits indicated a trend-level effect of groups (*p* = 0.069). (**D**) During the acute phase of treatment, the Penn Alcohol Craving Scale (PACS) scores showed a significant group–time interaction (*p* = 0.04). Craving levels of the active group were lower than those of the sham group by W2 at the trend level (*p* = 0.06) and significantly lower at W3 by the end of the acute treatment phase (*p* = 0.01; mean difference = 3, Cohen’s d = 0.48). (**E**) During the follow-up phase, PACS scores showed a trend-level main effect of groups (*p* = 0.07, mean difference = 3.7, Cohen’s d = 0.52). Data are presented as mean ± SEM in panels. * *p* < 0.05. BL, baseline.

**Figure 4 brainsci-13-01072-f004:**
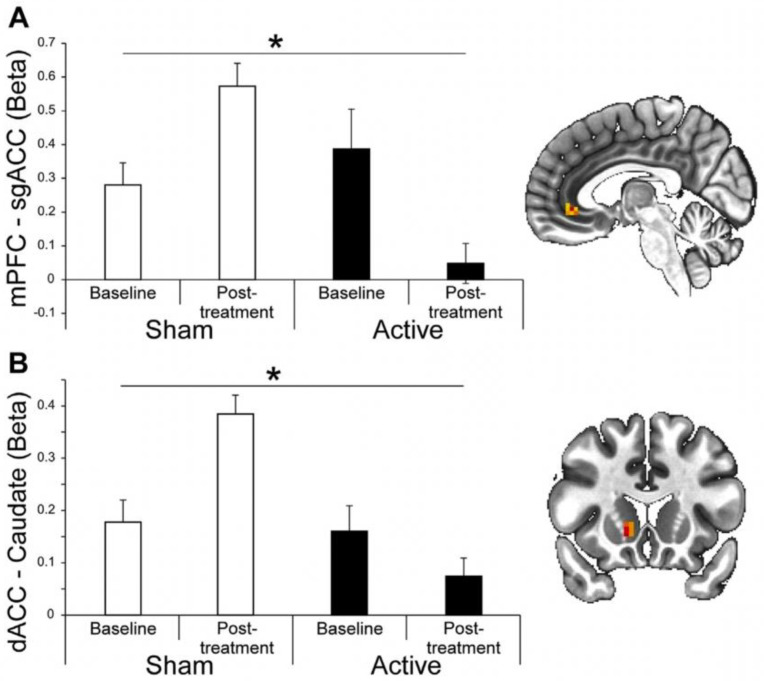
Adapted from [63]. Brain imaging results: Seed-based resting state functional connectivity findings at baseline and post-treatment. Significant group–time interaction for (**A**) mPFC–sgACC connectivity (*p* < 0.001) and (**B**) dACC–caudate connectivity were observed (*p* < 0.001). Significant main effects of groups for mPFC–dACC connectivity (*p* < 0.05) and dACC–caudate connectivity (*p* < 0.001). Data are presented as means ± SEM, * *p* < 0.05 between the active and sham groups.

## Data Availability

Not applicable.

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
