# Peer review of "Transcranial Magnetic Stimulation as a Tool to Promote Smoking Cessation and Decrease Drug and Alcohol Use"

_brainsci, 2023, doi:10.3390/brainsci13071072_

Round 1

Reviewer 1 Report

Thank you for writing this scientifically important, timely, and comprehensive review on TMS for the treatment of tobacco, alcohol, and cocaine use disorders. I had no major critiques as it was very well-written overall. However, I had a few minor comments below:

Page 1

- Consider incorporating other important SUDs (opioid use disorder, cannabis use disorder, etc.), particularly given the overdose epidemic in the US and the increase in cannabis use in the context of legalization.

- "studies using positron emission tomography (PET) [found?] reduced.."

Page 2

- extra "-" after third paragraph

- extra space after "lateral PFC" in fourth paragraph

Page 3

- end of second paragraph: "...relevant brain circuits, [and/which?] led [leading?] to clearance..."

Page 4

- line 17, "treatmens" misspelled

- same paragraph, last line: "immeadiately" misspelled

Figure 2:

- difficult to read the numbers on the scale for E [%]

Page 11

- first paragraph, second sentence "The" capitalized incorrectly

Page 18

- first paragraph, second-to-last sentence: "ACCor" misspelled

Page 24

- No section on TMS studies for other important SUDs (e.g. opioid and cannabis use disorders). Consider adding a section on TMS for other SUDs for which there is less available evidence, but similarly important. For example (not exhaustive):

               -Shen, Y., Cao, X., Tan, T., Shan, C., Wang, Y., Pan, J., He, H., & Yuan, T. F. (2016). 10-Hz Repetitive Transcranial Magnetic Stimulation of the Left Dorsolateral Prefrontal Cortex Reduces Heroin Cue Craving in Long-Term Addicts. Biological psychiatry, 80(3), e13–e14. https://doi.org/10.1016/j.biopsych.2016.02.006

               - Sahlem, G. L., Caruso, M. A., Short, E. B., Fox, J. B., Sherman, B. J., Manett, A. J., Malcolm, R. J., George, M. S., & McRae-Clark, A. L. (2020). A case series exploring the effect of twenty sessions of repetitive transcranial magnetic stimulation (rTMS) on cannabis use and craving. Brain stimulation, 13(1), 265–266. https://doi.org/10.1016/j.brs.2019.09.014

- second paragraph, first sentence: "TMS efficacy in each [major/primary?] SUD category." As noted above, other important SUDs (e.g. OUD) have been left out.

Page 25

- second to last paragraph, second sentence: "...were reduced after daily treatment with [TMS?] and some of these effects lasted..."

Author Response

Thank you for writing this scientifically important, timely, and comprehensive review on TMS for the treatment of tobacco, alcohol, and cocaine use disorders. I had no major critiques as it was very well-written overall. However, I had a few minor comments below…

We sincerely appreciate this feedback and regret that the grammatical errors were present in the submitted version.  Thank you.   

These have been fixed and we have expanded the references to other substance use disorders, including the addition of several paragraphs devoted to Opiate use disorder and Cannabis Use Disorder.            We hope the readers may find it illuminating.    

Reviewer 2 Report

The paper by Harmelech et al., is a systematic review of the current literature dealing with TMS as a therapeutic tool in various addictive states (see Antonelli et al., 2021). The paper is sound and well balanced. Importantly, it starts providing a possible link with neurotrasmitters involved in the neural basis of its clinical effects (i.e. dopamine), a point frequently neglected, which is very important to underscore the comparison with mechanisms underlying the use of drugs currently employed in addiction. Indeed, a hypothesis-driven approach is preferable to the most common trial and error.

However, enthusiasm drops abruptly when Nutt et al (ref 6) is cited in this regard. This is a miscitation and should be removed. The original (and only) dopamine hypothesis of drug addiction is Melis et al., 2005 (attached) which presented the fundamental basic science data to support the hypodopaminergic state hypothesis, later observed in human studies. The hypothesis was periodically updated (Diana, 2011) with latest in 2021 (Sanna et al., 2021) and currently widely recognized as a plausible mechanism to explain reduction in craving and drug-intake after various TMS protocols. 

Another surprising part is the lack of comments on the dopamine role in the discussion. In this portion of the manuscript the authors should elaborate on the dopamine mechanisms involved in the clinical effects induced by TMS.

Suggestions: 

1) The authors should emphasize the bilaterality of the H-coils versus the figure-of-eight, which is monolateral. Addiction is a 'whole brain disease' and a bilateral stimulus should 'in principle' be more efficacious than a monolateral one. Please elaborate on this point.

2) TMS offers a potential 'drug-free' treatment. This point should be treated by the authors and inserted into the advantages provided by the technique.

Author Response

The paper by Harmelech et al., is a systematic review of the current literature dealing with TMS as a therapeutic tool in various addictive states (see Antonelli et al., 2021). The paper is sound and well balanced. Importantly, it starts providing a possible link with neurotransmitters involved in the neural basis of its clinical effects (i.e. dopamine), a point frequently neglected, which is very important to underscore the comparison with mechanisms underlying the use of drugs currently employed in addiction. Indeed, a hypothesis-driven approach is preferable to the most common trial and error.

However, enthusiasm drops abruptly when Nutt et al (ref 6) is cited in this regard. This is a miscitation and should be removed. The original (and only) dopamine hypothesis of drug addiction is Meliset al., 2005 (attached) which presented the fundamental basic science data to support the hypodopaminergic state hypothesis, later observed in human studies. The hypothesis was periodically updated (Diana, 2011) with latest in 2021 (Sanna et al., 2021) and currently widely recognized as a plausible mechanism to explain reduction in craving and drug-intake after various TMS protocols.

Thank you for this and for the paper that was attached.   We have made revisions to the framing of the introduction.

Another surprising part is the lack of comments on the dopamine role in the discussion. In this portion of the manuscript the authors should elaborate on the dopamine mechanisms involved in the clinical effects induced by TMS.

Thank you. We have now revised the introduction and added some references regarding dopamine. More germane to TMS however, we have developed the scientific narrative regarding the role of frontal-striatal circuits in addictive behavior.  We have also added reference to the bilaterality of the brain regions and unique capabilities of the H-coils as the only coils currently available to simulate the left and right side simultaneously.    We have also revised Figure 2 to include a detailed and attractive overview of the electric field topography and strength for the H4 coils which is the only coil currently FDA-cleared in the addiction space.  These were great suggestions.

Reviewer 3 Report

1.     As a review, the auther should not excessively cite the content of one study. e.g. a double-blind, random-ized, sham-controlled (RCT) multisite clinical trial (16)

2.     (4/13) “The patient is a 38-year-old man” is appearing suddenly in the text. Is this case from Reference 16? It is not recommended to describe a case using a large paragraph in this review. Please make it concise.

3.     Some nouns or phrases need to be explained in the text. e.g. “CO-completers analysis set” in Fig.1, “MIDA” in Fig.3.

4.     In Fig.4 legend, “FU1-FU5” should be indicated by specific follow-up time.

5.     The contents of “Summary table of TMS studies” in tables are suggested to be further simplified.

6.     There are too many words in section “Conclusion”. Is it “Discussion”? Please rewrite these two sections.

1.     The manuscript contains a few of spelling and grammatical errors, please carefully check these errors and modify them.

Author Response

  1. As a review, the author should not excessively cite the content of one study. e.g. a double-blind, randomized, sham-controlled (RCT) multisite clinical trial (16)

We respect the source of this comment and have increased the references of other important studies. We have provided a more clear justification for the discussion of this study however as it is the biggest TMS trial in the drug and alcohol use domain and is the basis for the only FDA indication of TMS for drug or alcohol use disorders.  

  1. (4/13) “The patient is a 38-year-old man” is appearing suddenly in the text. Is this case from Reference 16? It is not recommended to describe a case using a large paragraph in this review. Please make it concise.

We have removed this section.

  1. Some nouns or phrases need to be explained in the text. e.g.“CO-completers analysis set” in Fig.1, “MIDA” in Fig.3.

We have clarified these elements.  The analysis is presented in the full Intent to treat sample (IIT) as well as the Per Protocol group (which includes the indiviudals in the IIT sample that finished all TMS sessions).  The Per Protocol group is listed here as Completers (CO).   In Figure 3, MIDA has been removed.

  1. In Fig.4 legend, “FU1-FU5” should be indicated by specific follow-up time.

These have now been added.

  1. The contents of “Summary table of TMS studies” in tables are suggested to be further simplified.

We have moved these to Supplementary Data to conserve space.

  1. There are too many words in section “Conclusion”. Is it “Discussion”? Please rewrite these two sections

The language has been revised.

Round 2

Reviewer 2 Report

I appreciate the changes made, but the new refs have not been included into the reference list(!), in spite of what the authors state in their response.

An additional point: the authors should consider that the whole paper sounds as an advertising piece of work for the various H-coils. Perhaps the authors like to keep that way but they should consider that the 'persuasive' role of the paper does not gain from it. 

Author Response

Reviewer #2:     

Comments and Suggestions for Authors

I appreciate the changes made, but the new refs have not been included into the reference list(!), in spite of what the authors state in their response.

We apologize for this oversight in the reference list.   We have gone through it carefully now, included all new references, and improved the formatting. We also appreciated your inclusion of the Melis et al article in the first review.   That is an excellent piece of work which will be useful to us in the future.

An additional point: the authors should consider that the whole paper sounds as an advertising piece of work for the various H-coils. Perhaps the authors like to keep that way but they should consider that the 'persuasive' role of the paper does not gain from it.

In this revision, we have now gone back through the manuscript very carefully and made some changes to the language such that it comes off as scholarly as even-handed as possible.  These changes include removing a figure, rewriting the section on AUD (highlighting 2 very comparable studies  - one of which was done with the H7 coil and one of which was done with a Figure 8 coil), and revising the Discussion.  We hope that the audience will see that we put a tremendous amount of time and effort into referencing the volumes of studies that have been done with figure 8 coils.  This can be seen in the long list of references as well as the supplementary tables.   

Reviewer 3 Report

No more comments or suggestions.

Author Response

Thank you